METHODS

# Achieving Occam's razor: Deep learning for optimal model reduction

**Botond B. Antal[1], Anthony G. Chesebro[1], Helmut H. Strey[1,2], Lilianne R. Mujica-Parodi[1,2,3], Corey Weistuch[4] ***

**1** Department of Biomedical Engineering, Stony Brook University, Stony Brook, New York, United States of America, **2** Laufer Center for Physical and Quantitative Biology, Stony Brook University, Stony Brook, New York, United States of America, **3** Santa Fe Institute, Santa Fe, New Mexico, United States of America, **4** Department of Medical Physics, Memorial Sloan Kettering Cancer Center, New York, New York, United States of America

* weistucc@mskcc.org

**Data Availability Statement:** The source code and data used to produce the results and analyses presented in this manuscript are available at www.lcneuro.org/analytic/fixfit.

## Abstract

All fields of science depend on mathematical models. *Occam's razor* refers to the principle that good models should exclude parameters beyond those minimally required to describe the systems they represent. This is because redundancy can lead to incorrect estimates of model parameters from data, and thus inaccurate or ambiguous conclusions. Here, we show how deep learning can be powerfully leveraged to apply Occam's razor to model parameters. Our method, FixFit, uses a feedforward deep neural network with a bottleneck layer to characterize and predict the behavior of a given model from its input parameters. FixFit has three major benefits. First, it provides a metric to quantify the original model's degree of complexity. Second, it allows for the unique fitting of data. Third, it provides an unbiased way to discriminate between experimental hypotheses that add value versus those that do not. In three use cases, we demonstrate the broad applicability of this method across scientific domains. To validate the method using a known system, we apply FixFit to recover known composite parameters for the Kepler orbit model and a dynamic model of blood glucose regulation. In the latter, we demonstrate the ability to fit the latent parameters to real data. To illustrate how the method can be applied to less well-established fields, we use it to identify parameters for a multi-scale brain model and reduce the search space for viable candidate mechanisms.

## Author summary

Mathematical modeling is a pillar of scientific inquiry, bridging the gap between theory and experimental observations. However, in complex systems such as those pervasive in biology (e.g., gene regulatory networks, multi-scale brain interactions, and drug pharma-cokinetics), different mechanisms can yield equally plausible explanations for the data. This ambiguity is not due to data limitations but rather to the equations that govern these systems. Both the interpretation and estimation of the parameters of these models are hindered by these intrinsic degeneracies. For this reason, we present a general tool that

**Funding:** This work was funded by the National Science Foundation (NSFNCS-FR 1926781) received by LRMP, by the Baszucki Brain Research Fund received by LRMP, by the National Institute of Health (T32-GM008444) received by AGC, and by the Marie-Josée Kravis Fellowship received by CW. The funders had no role in study design, data collection and analysis, decision to publish, or preparation of the manuscript.

**Competing interests:** The authors have declared that no competing interests exist.

harnesses the power of deep learning to automatically identify and rectify these ambiguities, allowing a broader range of models to be precisely determined by experimental data. We show, for example, that our method provides novel insights into the features of multi-scale brain dynamics that can be learned from functional neuroimaging. This represents a crucial initial step toward characterizing the mechanistic insights that different types of experiments can provide.

## 1 Introduction

Mathematical models are commonly used to describe the dynamical behavior of physical systems. Yet model parameters are not mere mathematical descriptions. The accurate estimation of parameter values can often yield deep mechanistic insight, whether with respect to the properties of particles [1], interactions among genetic networks [2], or the generation of neuronal signaling [3].

A fundamental challenge in parameter fitting and the construction of models stems from parameter redundancies [4, 5]. Parameter degeneracy is particularly problematic in multi-scale models, where emergent measured values exist many layers above their mechanistic parameters. This can lead to many different combinations of parameters fitting the observed data equally well regardless of the amount of data or the extent of observational noise, a phenomenon described as *overdetermined* or "sloppy models" [6–8]. Finding these non-unique solutions is also difficult, as parameter interactions can introduce numerous local minima that hinder algorithms for data fitting [9]. Consequently, there is often a trade-off between using an easier-to-interpret model with fewer parameters that fails to describe the system accurately and using a highly-detailed model that risks redundancy of its parameters [10].

A solution to this trade-off is to identify and account for parameter dependencies. One of the most widely-used tools to quantify interactions among parameters is the Fisher Information Matrix, which can be acquired through a nonlinear least-square Levenberg-Marquardt algorithm. This procedure, for given data, finds the locally best-fitting parameter values and their covariance matrix [8, 11]. In overdetermined models, these covariances are strong for pairs of parameters that are not separable. While the Fisher Information Matrix only characterizes linear interactions, other methods can also uncover nonlinear interactions [7, 12]. For example, one can determine parameter interactions by approximating the model's behavior as a function of its parameters using the Taylor series expansion [13]. Nevertheless, a common limitation of these methods is that they do not provide a functional form for the redundancies, and therefore cannot guide the parameter fitting process. A sensible strategy is to combine the redundant parameters into composite parameters with a unique best fit. Although suitable methods exist, they rely on studying the model analytically or through its local behaviors and derivatives [14], a strategy that would not be feasible for highly complex models. Thus, there remains a need for a more general approach.

Here we show how one may identify and estimate the largest set of lower-dimensional latent parameters uniquely resolved by model outputs for a given model (Fig 1). This reduced set of parameters can then be inferred from the data. The framework, *FixFit*, builds on previously established methods and consists of three steps (S1 Fig). First, we find these latent parameters using a neural network with a bottleneck [15] (Fig 1). We optimize the same architecture at variable bottleneck widths to identify the optimal latent dimension [16–18]. The ability of neural networks to approximate any function allows our method to be applied to models of arbitrary complexity [19–21]. Next, FixFit establishes the relationship between latent

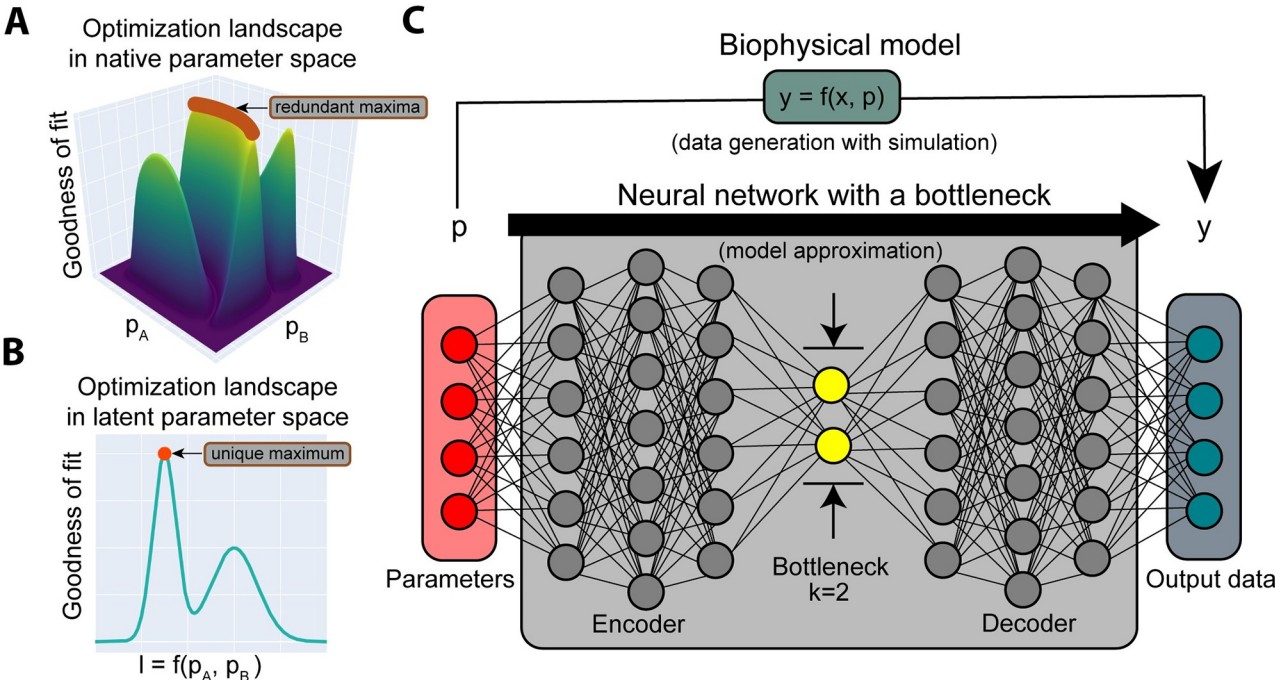

**Fig 1. FixFit compresses interacting parameters into a latent representation that can be uniquely inferred from data. A**: A schematic representing the goodness of fit landscape of a model with two interacting parameters. These interactions cause multiple parameter combinations to fit experimental data equally (the redundant maxima in red). **B**: The same landscape but with the two interacting parameters first combined into a single latent variable. In contrast to the native parameters, numerical fitting over latent variables will converge to a unique solution. **C**: FixFit generates such unique latent representations using a neural network with an encoder, bottleneck layer, and a decoder. After determining the optimal number ($k$) of latent (bottleneck) nodes (see Methods), the neural network is trained on pairs of parameters and their corresponding outputs from computational simulations of a model of interest. Following training, the bottleneck layer will include a representation of input parameters that is uniquely inferable from noiseless output data. With the latent representation established, the decoder section of the neural network can be combined with an optimizer to infer parameters in latent space from previously unseen output data. In addition, the encoder part can be combined with sensitivity analysis to determine the influence of input parameters on the latent representation. This enables us to characterize changes in different output samples in terms of underlying parameters that would not otherwise be accessible.

and original parameters using global sensitivity analysis [22]. Finally, we fit latent parameters to data using global optimization techniques [23].

Here we provide three use cases for FixFit. To establish validity against known systems, we first demonstrate its ability to recover nonlinear parameter combinations for the Kepler orbit model [24] and for a dynamic systems model of blood glucose regulation. Subsequently, to demonstrate its potential for scientific discovery, we then use FixFit to identify previously undiscovered parameter redundancies in the multi-scale Larter-Breakspear neural mass model [25, 26].

## 2 Results

### 2.1 Recovering the known parameter redundancies of the Kepler orbit model

The Kepler model [24] describes the elliptical orbit (pairs of angles ($\theta$) and radii ($r$), see Fig 2A) of two gravitationally-attracting bodies as a function of four input parameters ($m_1$, $m_2$, $r_0$, $\omega_0$) (see Methods). Ellipses, however, can be entirely described by two composite shape parameters, eccentricity ($e$) and the semi-latus rectum ($l$). As the dependencies between the four

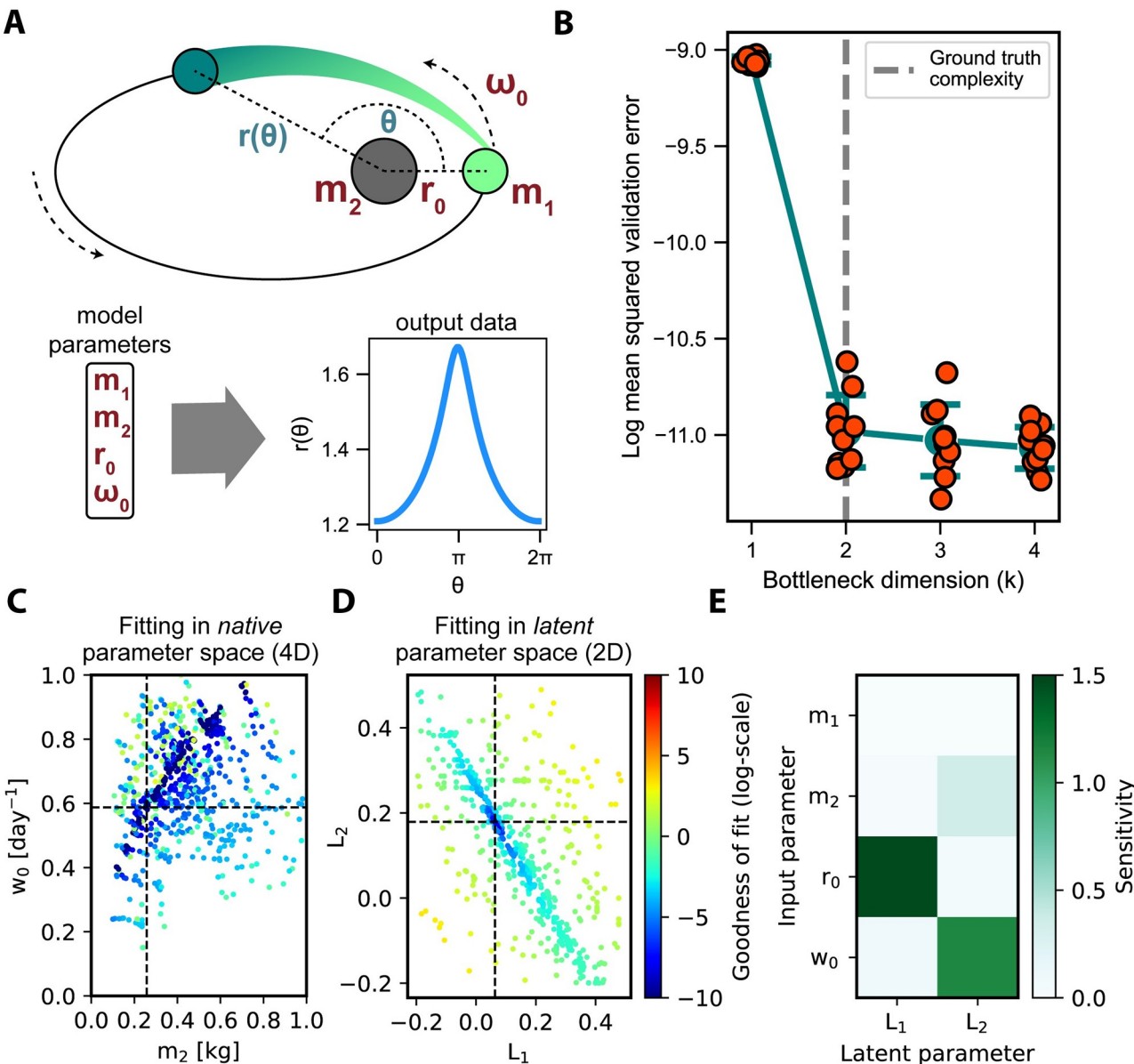

**Fig 2. Recovering the known parameter redundancies of the Kepler orbit model. A**: Diagram of an example Keplerian orbit, the corresponding input parameters (red), and the model outputs ($\theta$, $r$ pairs). **B**: Validation error of the compressed representation found by FixFit as a function of the bottleneck dimension ($k$). Multiple replicates were performed at each $k$ using a stochastic optimizer. Shown for each $k$ are the individual data points, mean, and standard error. FixFit correctly identified the underlying redundancy of the Kepler orbit model by indicating saturating error at $k$ = 2 (see Methods). In subsequent panels, we utilized one of the fitted neural network replicates at $k$ = 2. **C**: Objective landscape of two of the original four dimensions. Each point represents a function evaluation of the optimizer and the corresponding objective value (log-transformed sum of squares error) with respect to two parameters, $m_2$ and $\omega_0$ (scaled between 0 and 1), while fitting the model to a data sample. Dark blue points correspond to the global optima identified by the optimizer. The broad distribution of low error evaluations suggests a parameter redundancy. Consequently, the underlying ground truth parameters (marked by dashed lines) could not be uniquely identified by the optimizer. **D**: Objective landscape of the two latent dimensions. In contrast to the previous case, the same optimizer procedure run on the same sample converged to the correct and unique minimum in the latent parameter space ($L_1$, $L_2$) identified by FixFit. **E**: Structural and Correlative Sensitivity Analysis (SCSA) global sensitivities of the latent parameters to the four original parameters. Higher values of sensitivity (green) indicate a stronger influence. Considering the closed-form solution as a reference, SCSA correctly identified that parameter $m_1$ had no influence on outputs and that the remaining three parameters $m_2$, $r_0$, and $\omega_0$ together determined the two latent parameters.

input and two composite parameters are known analytically, our results can be compared against a ground truth.

After generating simulated data using the original model, we trained a neural network with a variable-width bottleneck layer ($k = 1...4$, S2 Fig) to determine, from simulated data, the underlying complexity of the Kepler model (see Methods). The validation error saturated at $k = 2$, suggesting that the underlying minimal representation, consistent with the ground truth, is two-dimensional (Fig 2B). In subsequent analyses, we thus utilized a trained neural network from one of the replicates at $k = 2$ to encode a minimal representation of the four input parameters.

To demonstrate the practical issue of inferring degenerate model parameters, we attempted to uniquely infer all four original parameters from a simulated Kepler orbit from the training sample (see Methods). As expected, the global optimizer found a wide set of equally optimal solutions across the parameter space (Fig 2C). By contrast, global optimization on the same sample but in latent parameter space (through the decoder section of the neural network, see Methods) recovered a unique solution that coincided with the expected ground truth (Fig 2D).

Next, we demonstrate that, by employing Structural and Correlative Sensitivity Analysis (SCSA), FixFit can quantify relationships between the original and latent model parameters (see Methods). Applied to the Kepler model (Fig 2E), the approach provides global sensitivities consistent with closed-form solutions for the two intermediate terms $e$ and $l$. Firstly, as expected, no sensitivity was detected with respect to $m_1$. As such, $m_1$ was a completely redundant parameter. The remaining three parameters, $r_0$, $m_2$ and $\omega_0$ together determined $L_1$. Finally, $r_0$ alone influenced $L_2$. Both ground truth terms, $e$ and $l$, by contrast, included all three of these parameters (Eqs 3 and 4). However, these analytical expressions share a common term with respect to the three input parameters and differ only by an additional power of $r_0$ in $l$. By exploiting this relationship, FixFit separates the influence of $r_0$ from that of $m_2$ and $\omega_0$, thus providing a sparser latent representation compared to the ground truth.

## 2.2 Characterizing parameter degeneracies in a dynamic system model of blood glucose-insulin regulation

The $\beta IG$ model of glucose-insulin regulation is a well-established nonlinear model describing blood glucose dynamics in response to glucose uptake [27, 28]. Six parameters ($C$, $S_i$, $p$, $\alpha$, $\gamma$, $u_{ext}$) govern the regulation among three state variables, which are glucose, insulin and $\beta$-cell function (Fig 3A) (see Methods). As typically we can only observe glucose levels, the effects of certain parameters become indistinguishable from each other. Specifically, it has been previously shown that parameters $S_i$ and $p$ are not separable; only their product is identifiable from glucose time-series alone [27]. Additional redundancies are anticipated due to the simplistic representation of glucose uptake as impulse functions at fixed time intervals. Here, we illustrate the ability of our method to characterize parameter degeneracies from time-series data and showcase how the learned representation can be fitted to experimental data.

After simulating a large set of time-series using the original model and evenly sampled input parameters, we trained a neural network with varying bottleneck widths on model input-output data pairs (S3 Fig). Notably, we observed minimal error starting at $k = 4$ nodes in the bottleneck indicating the presence of two degrees of redundancy among the six model parameters (Fig 3B). These findings were supported by our supplementary analysis of structural identifiability which also revealed four uniquely observable parameters for the same model (see Methods). Consequently, we again utilized a trained network from one of the $k = 4$ replicates for downstream analyses.

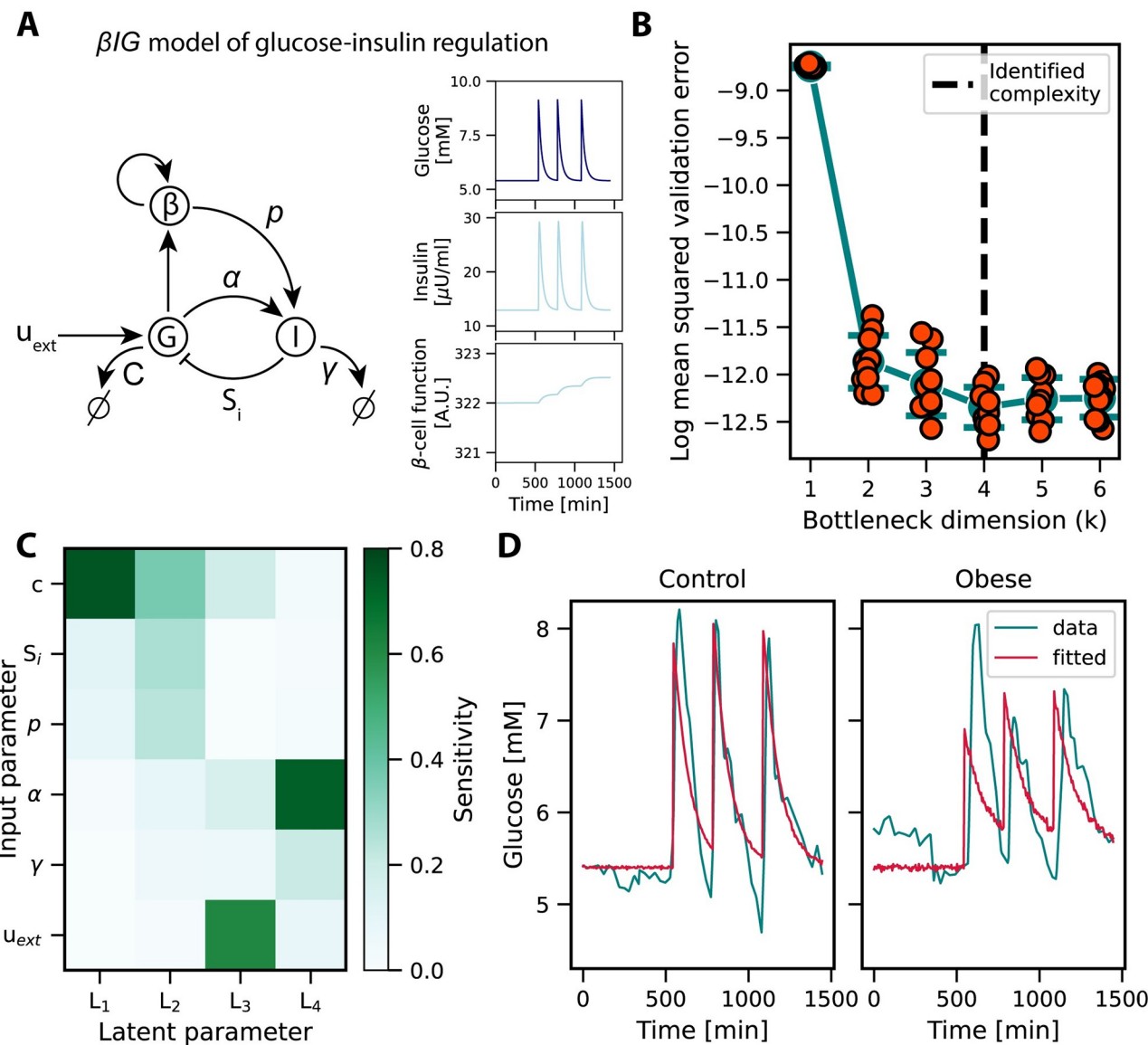

**Fig 3. Characterizing parameter degeneracies in a dynamic system model of blood glucose-insulin regulation. A**: Diagram of the *βIG* model of glucose-insulin regulation. The dynamic interactions among three state variables—glucose (G) and insulin (I) blood concentrations, and *β*-cell mass (*β*) —are determined by six parameters, including an external input $u_{ext}$. The two null signs indicate degradation processes. Representative dynamics for the three variables are shown on the right, with glucose concentration highlighted as the only observable variable in our experiments. **B**: The validation error of the neural network reached its minimum when the bottleneck dimensionality was set to $k = 4$ nodes, indicating that four distinct parameters could be resolved from the model outputs. Shown for each k are the individual data points, mean, and standard error. **C**: Global sensitivity analysis results of the four latent parameters with respect to the original model parameters. The sensitivities are color-coded. All parameters appeared to influence model outputs. The ground truth redundancy among $S_i$ and $p$ is apparent, with another redundancy shared across the model parameters. **D**: The learned representation was successfully fitted to measured data using a global optimizer. Glucose time-series data obtained from both healthy and obese individuals are shown in green, alongside model predictions in red.

We investigated the detected parameter redundancies by transforming the original input parameters into latent parameters using the encoder portion of the trained neural network. Subsequently, we conducted a global sensitivity analysis on the latent parameters relative to the original ones. Our results revealed that all six parameters influenced model behavior, thus the indicated redundancies were shared across multiple parameters (Fig 3C). Consistent with

prior research, our global sensitivity analysis confirmed the inseparability of $S_i$ and $p$, as they exhibited influence on the same latent parameters. The additional degree of redundancy appeared to be shared across several parameters, bringing the total redundancy count to four.

Next, we utilized a global optimizer to fit the learned latent representation to measured time-series (see Methods) from human subjects, comprising one dataset from a healthy control and another from an obese individual [29]. Our analysis yielded closely fitting solutions for both cases, elucidating key features (Fig 3D). Notably, we observed impaired dynamics in the obese subject, presumably attributed to diminished insulin sensitivity, leading to a slower rate of glucose clearance from the bloodstream. While certain characteristics remained unaccounted for, such as an undershoot in glucose concentration and an elevated glucose peak following fasting in the obese case, these discrepancies likely stem from limitations of the original model rather than a failure to fit the representation found by FixFit. For example, the maximum decrease between the first and second peak heights observed across all simulations was 0.1%, suggesting that the model cannot capture this feature observed in obese individuals.

## 2.3 Identifying novel parameter relationships in a multi-scale brain model

The Larter-Breakspear model is commonly used to connect microscopic neuronal properties with emergent brain activity, such as that measured using functional magnetic resonance imaging (fMRI) [30] (Fig 4). This is achieved by modeling the voltage-gated ion dynamics (here, $Na^+$, $K^+$, and $Ca^{2+}$) of a population of neurons called a "neural mass" [25, 26, 31]. Previous studies have further shown that, by coupling multiple neural masses according to the inter-regional connectivities measured through Diffusion Tensor Imaging (DTI), one can simulate semi-realistic fMRI dynamics [32]. As a challenge, this model has many parameters such that even after assigning identical parameter values to all 78 of the resulting brain regions and fixing biologically inert parameters, the model still has eleven remaining parameters corresponding to coupled biological processes (see Methods). Furthermore, these simulated fMRI activities are typically studied using functional connectivity (FC) or the matrix of region-to-region Pearson correlations [33], thus resulting in a reduction of information. To address this issue, we applied FixFit to resolve potential redundancies and, more broadly, characterize new relationships among the parameters. Microscopic model parameters, such as ion channel conductivity, are often implicated in disease mechanisms and potential therapeutic targets, which makes their determination valuable [34].

As with the previous examples, we first trained neural networks with various bottleneck widths (see Methods) to quantify the information content of our fMRI simulations (S4 Fig). The minimum validation error occurred at $k = 4$ (Fig 5A), suggesting that four latent parameters ($L_1, L_2, L_3, L_4$) capture the composite effect of the original eleven parameters.

Using a representation acquired at $k = 4$, we next give an example of how to infer latent parameters from model outputs and interpret the results in input space. For this purpose, we first perturbed the $Na^+$ reversal potential ($V_{Na}$) and simulated new samples under two conditions ($V_{Na} = \{0.48, 0.54\}$) while keeping all other parameters fixed (Fig 5B). We observed that increasing $V_{Na}$, on average, reduced all correlations and introduced more anti-correlations across the brain. We then performed parameter inference in latent space in both conditions, and in each case, the global optimizer converged to a single unique solution. The two solutions differed only along the $L_1$ and $L_2$ axes, whereas detected changes with respect to $L_3$ and $L_4$ were minimal.

To interpret the observed shifts in the fitted latent parameters, we again applied global sensitivity analysis (Fig 5C). We found three parameters, $g_{Na}$, $a_{ee}$, and $r_{NMDA}$ had no effect on the latent parameters and therefore did not affect FC model outputs. Since no shifts in this

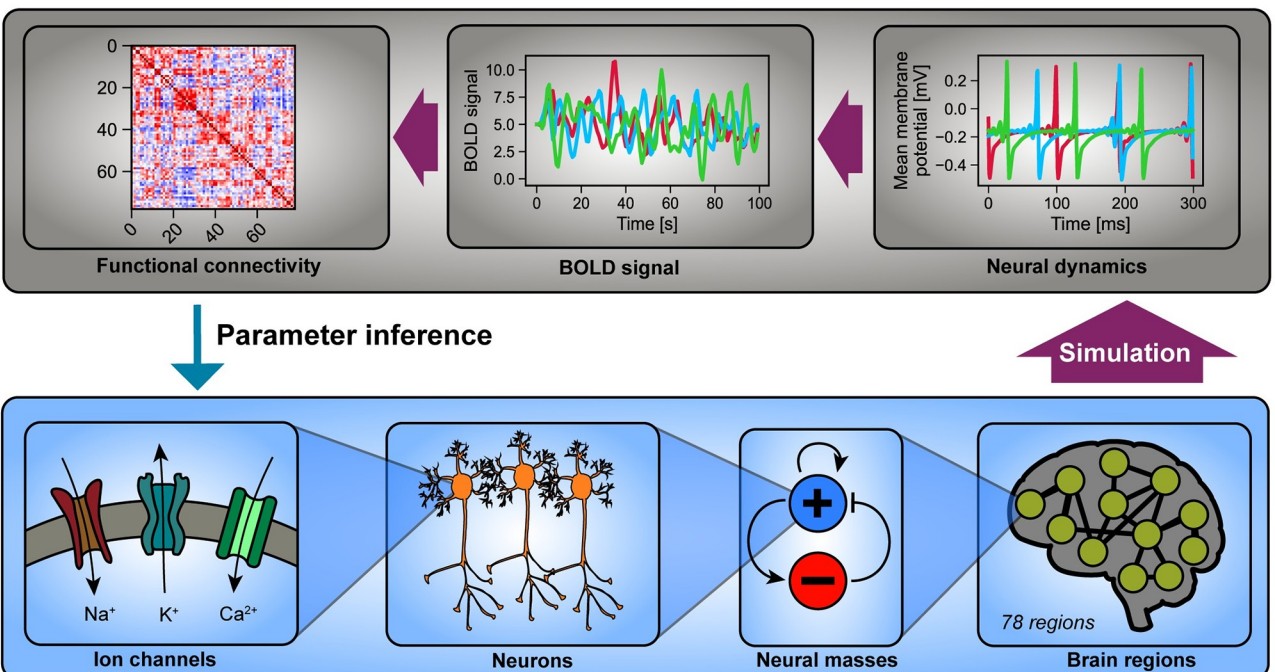

**Fig 4. The Larter-Breakspear brain model connects microscopic properties of neurons with patterns of brain-wide activity.** This model produces brain dynamics by considering neuronal properties on multiple spatial scales (bottom). On the bottom scale, it takes into account transmembrane ion transport (bottom left), which is the basis of neuronal dynamics. Ion transport is facilitated by voltage-gated ion channels that are separately specified for $K^+$, $Na^+$, and $Ca^{2+}$ ions. Next, a mean-field approximation aggregates single-cell behavior into the population level of neural masses. An interacting pair of an excitatory (+) and an inhibitory (-) population is integratively modeled to form individual brain regions. Finally, on the whole-brain scale, a network of 78 connected brain regions (connected based on diffusion tensor imaging) is considered to complete the spatial span of the model (bottom right). The simulated region-specific brain dynamics are then transformed to achieve compatibility with functional MRI, a non-invasive neuroimaging modality (top). Simulated time series (top right) are first converted into blood-oxygen-level dependent (BOLD) signals using a hemodynamic response function. Then, to remove phase-specific information, all-to-all Pearson correlations are computed among the time series resulting in a 78-by-78 functional connectivity (FC) matrix (top left).

example were observed in $L_3$ and $L_4$, 5/8 of the remaining parameters can be eliminated. This reduced the candidate parameters from eleven down to three, with $V_{Na}$ contributing to the shift in $L_1$ and $L_2$ (Fig 5C) (consistent with Fig 5B). As demonstrated by this example, FixFit can be used to narrow down potential mechanisms for the observed changes in latent parameters. This reduction makes the remaining uncertainty tractable to resolve in specialized experiments that target individual parameters on the single neuron scale.

## 3 Discussion

While many methods exist for fitting model parameters to data, most of them are limited to situations where there is a need for a single, optimal solution. However, many different parameter sets often fit the data equally well due to complex parameter dependencies and the limited information content of experimental measurements [14]. In this paper, we have presented FixFit, an approach for discovering compressed representations of models with redundant parameters. This allows for the identification of unique best-fit latent parameters sets in both simulated and real data. Our approach correctly identified that, as previously established, two latent parameters were sufficient to characterize the Kepler orbit model. In another model describing physiological control of blood glucose, we reproduced that parameters describing insulin production and insulin sensitivity were inseparable based on measured glucose levels

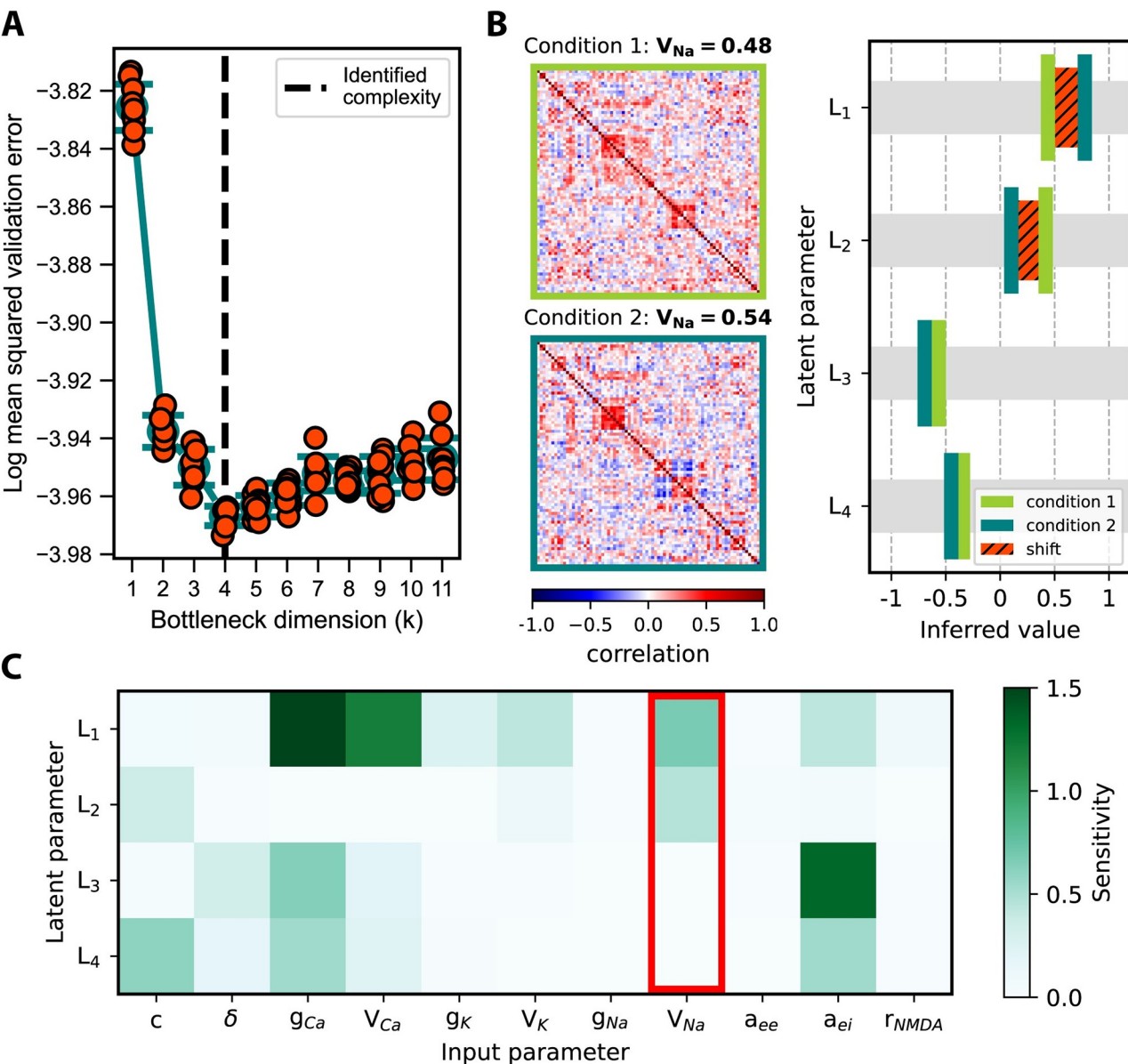

**Fig 5. Identifying novel parameter relationships in a multi-scale brain model. A**: Validation error of the Larter-Breakspear model as a function of the bottleneck dimension ($k$). Shown for each $k$ are the individual data points, mean, and standard error. The minimal error occurs at $k = 4$, implying that the eleven model parameters can be summarized by four latent parameters while still uniquely mapping to output space. **B**: Simulated functional connectivity matrices (left) and corresponding fitting results in latent space (right) for two different values of $V_{Na}$. Fitting (see Methods) converged to global minima in both conditions and indicated shifts in $L_1$ and $L_2$ (right, orange). **C**: Global sensitivities, as computed through SCSA, of the four latent parameters to the 11 original model parameters. Latent parameter sensitivities to input parameters are color-coded. Three input parameters, $g_{Na}$, $a_{ee}$, and $r_{NMDA}$, do not appear to influence model outputs. $V_{Na}$ (marked with a red bracket), which we investigated in our perturbation experiment, influenced latent parameters $L_1$ and $L_2$ but not $L_3$ and $L_4$. These sensitivities are consistent with our fitting results in panel B, where we only detected shifts in $L_1$ and $L_2$.

alone. Compressing these parameters enabled fitting to be performed on real data. Furthermore, we demonstrated in the Larter-Breakspear brain network model that we can use changes in such composite features to narrow down parameter candidates that shift in response to experimental perturbations, enabling FixFit to assess the features that specific types experiments can and cannot resolve.

The purpose of FixFit significantly differs from traditional parameter estimation methods. As described earlier, conventional fitting approaches often struggle with models with structural redundancies. For example, regular and variational Bayesian methods cannot appropriately sample or approximate flat posteriors, such as those resulting from structural redundancies handled by FixFit [12]. In contrast, FixFit does not directly perform parameter estimation. While we used FixFit with a particular optimizer here, users have the flexibility to choose any parameter-fitting strategy, including a Bayesian approach, according to their preferences. Thus, FixFit is designed to identify a reduced, redundancy-free model. The new reduced model can then be fitted using standard parameter fitting strategies.

Care must be taken when applying FixFit to systems with observational and model noise. Observational noise affects the certainty with which model parameters can be inferred from data but not structural parameter redundancies [35, 36]. Thus, to handle observational noise, FixFit should first be applied to noise-free simulated data to find the optimal parameter representation, then use objective-based or Bayesian inference methods for latent parameter estimation. Model noise, such as in Langevin dynamics, is intrinsic to the system dynamics and can affect parameter redundancies and estimation [37, 38]. Our method could then be extended by utilizing either feedforward [39] or recurrent neural networks [40] that have the ability to learn stochastic dynamics. By keeping the noise parameters fixed in each application of FixFit, the generalized approach could potentially be used to gauge how inferable complexity depends on the type and amount of intrinsic model noise.

As with other applications of neural networks, the practical choice of architectures requires trial and error. However, the freedom to add, in theory, unlimited additional training samples means that FixFit can be utilized on any neural network architecture capable of expressing the mathematical relationship between model inputs and outputs with a restricted number of latent dimensions chosen by the user [15]. For our current examples, we also employed multi-layer feedforward neural networks with variable layer sizes between the input, bottleneck, and output layers, offering numerous opportunities for compression and decompression.

As with many other dimensionality reduction methods [41, 42], the latent representation provided by FixFit is not unique. Future generalizations of the method could exploit this additional layer of flexibility to select, among equally accurate alternatives, the representation where latent variables are functions of the fewest possible input parameters. This would be done by adding additional sparsity constraints, such as variance maximization (Varimax), during neural network training and architecture selection [41, 43].

The idea of achieving a minimal and sufficient representation of a physical system by implementing compression on dynamical systems has been previously explored [44]. Our approach builds upon this notion by linking the compressed representation to model parameters, which enables us to address redundancies in the original parameter space. However, to ensure the utility of this representation discovered by our tool, it must be interpreted in relation to the input parameters. Various methods are available to interpret the components of an already-trained neural network. These include sensitivity analysis [45], feature importance scoring [46], activation maximization [47], and gradient-based saliency mapping [48]. For the demonstration of our framework, we used SCSA, a sensitivity analysis variant, to quantify the impact of each model parameter on the latent parameters [22]. However, as with the other mentioned approaches, sensitivity analysis does not provide explicit formulas for the nonlinear parameter relationships identified by the neural network. Besides revealing more information about relationships between input parameters, such formulas would allow latent parameters to be understood as composite parameters that may be easier to interpret (i.e., eccentricity in the Kepler orbit model). This, in turn, could allow FixFit to develop intuitive, simplified models. Symbolic

regression offers a possible avenue to achieve this by approximating the encoder [49, 50] or by itself being incorporated into the neural network [51, 52].

Future studies could utilize our framework to rank different experimental designs based on how well they resolve specific parameters of interest in simulations. In neuroscience, for example, brain-wide activities can be measured in a multitude of ways, including by fMRI, electroencephalography (EEG), and magnetoencephalography (MEG). FixFit could use simulations of these modalities from models (e.g., Larter-Breakspear) to determine which parameters each can resolve and, as a result, when to use them. Further, our framework can suggest targeted follow-up experiments to augment pre-existing data and inferred latent parameters. Related methods of optimal design aim to select experimental conditions such as temperatures, concentrations, and sample sizes to most easily resolve the parameters of a fixed model [53–55]. Thus, our tool could work synergistically with these prior approaches to optimize future data collection.

## 4 Conclusion

In conclusion, we present FixFit, a method to identify unique representations of redundant model parameters of computational models. Our tool enables scientists to select the most informative measurement modalities for their experiments, particularly in fields such as neuroscience, where it is still ambiguous what information experiments can resolve. Additionally, using our approach, researchers can utilize an extensive library of existing computational models for parameter inferences. These efforts can be further enhanced by specialized experiments that address the remaining parameter uncertainties, leading to greater synergy between modeling and experimentation.

## 5 Methods

### 5.1 Models

**5.1.1 Kepler orbit model.** The classical Kepler orbit model captures the gravitational motion of two orbiting planetary bodies through four input parameters (Fig 2A) [24]: the mass of the orbiting body ($m_1$), the mass of the body fixed at the focus point ($m_2$), the closest distance between the two bodies ($r_0$) and the corresponding initial angular velocity ($\omega_0$). To facilitate later training of a neural network, the units of each parameter were chosen such that the parameter values had comparable magnitudes. Thus, $m_1$ and $m_2$ were expressed in $kg$, $r_0$ in $m$, and $w_0$ in $day^{-1}$. The center of mass of the orbiting body evolves according to:

$$m_1 \frac{d^2 r}{dt^2} - m_1 r \omega^2 = \frac{G m_1 m_2}{r^2} \tag{1}$$

where $G$ is the universal constant of gravitation (approximately $0.5 m^3 kg^{-1} day^{-2}$), $r$ is the distance between the two bodies, and $\omega$ is the angular velocity $\frac{d\theta}{dt}$. As is apparent, the mass of the orbiting body ($m_1$) cancels in the above equation and thus cannot be resolved from data. More broadly, the dynamics are highly constrained by the conservation of energy and angular momentum. As a result, the orbit $r$ can be readily solved as a function of the angle $\theta \in [0, 2\pi)$ swept by the orbiting object (starting at $\theta = 0$) [24]. These solved orbits are simple ellipses and thus can be described using only two pieces of information, the eccentricity $e$ (the shape) and the semi-latus rectum $l$ (the size), as follows:

$$r(\theta) = \frac{l}{1 + e \cos(\theta)} \tag{2}$$

with

$$e = \left| \frac{r_0^3 \omega_0^2}{Gm_2} - 1 \right| \tag{3}$$

$$l = \frac{r_0^4 \omega_0^2}{Gm_2} \tag{4}$$

The above two equations establish a ground truth for the degeneracy that we aimed to characterize and overcome during parameter inference. Specifically, parameter $m_1$ is canceled out and is therefore a completely redundant parameter. The remaining three parameters ($m_2$, $r_0$, $\omega_0$) can be compressed into two terms and still uniquely map to outputs [24].

**5.1.2 $\beta IG$ model of glucose-insulin regulation.** The $\beta IG$ model of glucose insulin regulation (Fig 3A) is a nonlinear model of dynamical compensation that has been described in previous work [27, 28]. To briefly summarize other work in this area, the concentration of glucose ($G(t)$), with endogenous ($u_0$) and exogenous ($u(t)$) input sources, is regulated by insulin ($I(t)$) and provides feedback to stimulate the growth of pancreatic $\beta$-cells ($\beta(t)$). The system is governed by the equations:

$$\frac{dG}{dt} = u_0 + u(t) - (C + S_i I)G \tag{5}$$

$$\frac{dI}{dt} = p\beta \frac{G^2}{\alpha^2 + G^2} - \gamma I \tag{6}$$

$$\frac{d\beta}{dt} = \beta(\lambda_+(G) - \lambda_-(G)) \tag{7}$$

where $\lambda_+$, $\lambda_-$ are nonlinear functions of the production and removal rate of $\beta$-cells, respectively, given by

$$\lambda_+(G) = \frac{\mu_+}{1 + \left(\frac{8.4}{G}\right)^{1.7}} \tag{8}$$

$$\lambda_-(G) = \frac{\mu_-}{1 + \left(\frac{G}{4.8}\right)^{8.5}} \tag{9}$$

Prior work has shown the parameters $S_i$ and $p$ are nonidentifiable when only the glucose time series $G(t)$ is observed [27]. In parallel with applying FixFit, we checked parameter redundancies using the StructuralIdentifiability.jl package in Julia, which allows for the assessment of identifiable parameters given different observation functions [56]. When observing only the glucose time series, this package indicated the total number of identifiable parameters was four, with the meal bolus $u_{ext}$ and insulin secretion rate due to glucose $\alpha$ not being globally identifiable. The reparameterization to a minimal latent space is provided in the code for this section as well.

**5.1.3 Multi-scale brain model.** The Larter-Breakspear model (Fig 4) describes the dynamics of a group of coupled brain regions, each modeled as a population of neurons [25, 26, 31, 32]. Each brain region captures the averaged synaptic processes and voltage-dependent ion transport ($K^+$, $Na^+$, and $Ca^{2+}$) of its constituent neurons. These effective dynamics are described through three state variables: mean excitatory membrane voltage V(t), mean inhibitory membrane voltage Z(t), and the proportion of open potassium channels W(t). Note that

although $V$, $Z$, and $W$ depend on time, this notation is omitted in the following equations for clarity. These states of each $i$th brain region evolve according to:

$$
\begin{aligned}
\frac{dV_i}{dt} = \\
-(g_{\text{Ca}} + r_{\text{NMDA}} a_{ee}[(1-c)Q_V + cQ_i^{\text{network}}])m_{\text{Ca}}(V_i - V_{\text{Ca}}) \\
-(g_{\text{Na}} m_{\text{Na}} + a_{ee}[(1-c)Q_V + cQ_i^{\text{network}}])(V_i - V_{\text{Na}}) \\
-g_{\text{K}} W_i(V_i - V_{\text{K}}) - g_L(V_i - V_L) - a_{ie} Z_i Q_z + a_{ne} I_0
\end{aligned}
\tag{10}
$$

$$
\frac{dZ_i}{dt} = b(a_{ni} I_0 + a_{ei} V_i Q_V)
\tag{11}
$$

$$
\frac{dW_i}{dt} = \phi \frac{m_{\text{K}} - W_i}{\tau_{\text{K}}}
\tag{12}
$$

Where $Q_V/Q_Z$ are excitatory/inhibitory mean firing rates, and $m_{\text{Na}}$, $m_{\text{K}}$, $m_{\text{Ca}}$ are ion channel gating functions. These are computed as:

$$
Q_V = 0.5 Q_{V_{\text{max}}}(1 + \tanh(\frac{V - V_T}{\delta}))
\tag{13}
$$

$$
Q_Z = 0.5 Q_{Z_{\text{max}}}(1 + \tanh(\frac{V - V_Z}{\delta}))
\tag{14}
$$

$$
m_{\text{ion}} = 0.5(1 + \tanh(\frac{V - T_{\text{ion}}}{\delta_{\text{ion}}}))
\tag{15}
$$

Brain regions are connected through the coupling term $Q_i^{\text{network}}$, which is scaled with a global coupling constant $c$, to produce whole-brain-scale dynamics. $Q_i^{\text{network}}$ is given by:

$$
Q_i^{\text{network}} = \frac{\sum_j u_{i,j} Q_{V_j}}{\sum u_{i,j}}
\tag{16}
$$

We considered 78 brain regions selected from the Desikan-Killiany atlas included in FreeSurfer, with inter-regional structural connectivity ($u_{i,j}$) that was determined from diffusion tensor imaging (DTI) data from 13 healthy human adults [32, 57]. All parameters of the Larter-Breakspear model are described in Table 1.

Next, to produce a signal compatible with functional MRI the simulated region-specific excitatory signals were transformed into blood-oxygen-level-dependent (BOLD) signal via the Balloon-Windkessel model with standard parameters [32, 58]. Finally, we derived functional connectivity (FC) from a simulated BOLD signal to yield a phase-invariant signal. We quantified FC with all-to-all Pearson correlation coefficients among the 78 regions, resulting in a 78-by-78 correlation matrix for each simulation.

## 5.2 Data generation

**5.2.1 Kepler orbit model.** Training and validation data for the neural network were generated using Eqs 2–4. The four input parameters, $m_1$, $m_2$, $r_0$, and $\omega_0$, were sampled with a four dimensional Sobol sequence (SciPy v1.7.1 [23]). All four parameters were drawn from a range of [0.1, 1] (S5 Fig). Next, a subset of samples was rejected based on an eccentricity criterion. A raw parameter set was discarded if $e > 0.95$ or $e < 0.7$. These two conditions ensured all

**Table 1. Parameters of the Larter-Breakspear multi-scale brain model.**

| Parameter | Description |
| --- | --- |
| $V_{Na}$ | Na$^+$ reversal potential |
| $V_K$ | K$^+$ reversal potential |
| $V_{Ca}$ | Ca$^{2+}$ reversal potential |
| $V_L$ | Leak channels reversal potential |
| $g_{Na}$ | Na$^+$ conductance |
| $g_K$ | K$^+$ conductance |
| $g_{Ca}$ | Ca$^{2+}$ conductance |
| $g_L$ | Leak channels conductance |
| $T_{Na}$ | Na$^+$ channel threshold |
| $T_K$ | K$^+$ channel threshold |
| $T_{Ca}$ | Ca$^{2+}$ channel threshold |
| $\delta_{Na}$ | Na$^+$ channel threshold variance |
| $\delta_K$ | K$^+$ channel threshold variance |
| $\delta_{Ca}$ | Ca$^{2+}$ channel threshold variance |
| $V_T$ | Excitatory neuron threshold voltage |
| $Z_T$ | Inhibitory neuron threshold voltage |
| $\delta$ | Variance of thresholds |
| $Q_{V_{max}}$ | Excitatory population maximum firing rate |
| $Q_{Z_{max}}$ | Inhibitory population maximum firing rate |
| $a_{ee}$ | Excitatory-to-excitatory synaptic strength |
| $a_{ei}$ | Excitatory-to-inhibitory synaptic strength |
| $a_{ie}$ | Inhibitory-to-excitatory synaptic strength |
| $a_{ne}$ | Non-specific-to-excitatory synaptic strength |
| $a_{ni}$ | Non-specific-to-inhibitory synaptic strength |
| $I_0$ | Subcortical excitatory input |
| $b$ | Time scaling factor |
| $\phi$ | Temperature scaling factor |
| $\tau_K$ | K$^+$ relaxation time |
| $r_{NMDA}$ | NMDA/AMPA receptor ratio |
| $c$ | Global region-to-region coupling constant |

resultant orbits were ellipse-shaped with moderate eccentricity. The final sample sizes were 2,276 for training and 253 for validation. Output space consisted of $r(\theta)$ values computed at 100 evenly distributed $\theta$ values within the range of $[0, 2\pi]$ for each input parameter set. The computed $r(\theta)$ values were log-transformed to narrow down their range, thus ensuring a favorable output space for the neural network (S6 Fig).

**5.2.2 $\beta IG$ model of glucose-insulin regulation.** Following previous work [27], we simulated for a 24-hour time period with three meals of equal sizes at 9:00AM, 1:00PM, and 6:00PM. These were modeled as three Dirac delta functions at these times with a height of $u_{ext}$. The parameters of the $\beta IG$ model equations are given in Table 2, with their physiological explanations and standard values. For our simulations, we allowed five parameters ($C$, $S_i$, $p$, $\alpha$, $\gamma$) to vary freely by an order of magnitude above and below their standard value, the meal bolus ($u_{ext}$) to vary within normal physiological limits [59], and two parameters to remain fixed ($\mu_+$ and $\mu_-$) as they contribute <1% change per day (the length of our simulations). Values for the six parameters that varied were drawn from a Sobol sequence to ensure an even distribution within the sampling space. Initial conditions $G(0) = G_0$ and $\beta(0) = \beta_0$ were fixed to the original parameters

**Table 2. Parameters of the $\beta IG$ glucose-insulin regulation model.**

| Parameter | Description | Standard Value (sv) | Units | Variation |
|---|---|---|---|---|
| $C$ | Glucose removal rate at zero insulin | 1e-3 | $\frac{1}{min}$ | $(0.1, 10)\cdot sv$ |
| $S_i$ | Insulin sensitivity | 5e-4 | $\frac{ml}{\mu U\cdot min}$ | $(0.1, 10)\cdot sv$ |
| $p$ | Insulin secretion per cell | 3e-2 | $\frac{ml}{\mu U\cdot min}$ | $(0.1, 10)\cdot sv$ |
| $\alpha$ | Insulin secretion rate due to glucose stimulation | 7.85 | mM | $(0.1, 10)\cdot sv$ |
| $\gamma$ | Insulin removal rate | 3e-1 | $\frac{1}{min}$ | $(0.1, 10)\cdot sv$ |
| $u_{ext}$ | Meal intake glucose amount | 4 | mM | $(2, 5)$ |
| $\mu_+$ | Increase in beta cell functional mass | $\frac{0.021}{24\cdot60}$ | $\frac{1}{min}$ | $< 1\%$ change/day |
| $\mu_-$ | Decrease in beta cell functional mass | $\frac{0.025}{24\cdot60}$ | $\frac{1}{min}$ | $< 1\%$ change/day |

[27]. Initial conditions $I(0) = I_0$ and $u_0$ were set to ensure the system was at a steady state when initialized and were completely dependent on the other choices of parameters such that

$$I_0 = p\beta_0 \frac{G_0^2}{\gamma(\alpha^2 + G_0^2)} \tag{17}$$

$$u_0 = G_0(C + S_iI_0) \tag{18}$$

After the simulations, we applied a series of transformations to the data to expedite convergence for the neural network. First, the values for each input parameter were rescaled to the range of [0, 1] (S7 Fig). Next, the simulated time-series were also rescaled to the [0, 1] while preserving their relative scaling across different samples. Subsequently, the time-series were downsampled to time increments of 5 minutes (S8 Fig). In total, 6,984 samples were retained for training, and 1,647 samples were allocated for validation. The same transformations were applied to the measured time-series that we utilized to demonstrate model fitting.

**5.2.3 Multi-scale brain model.** We first applied domain knowledge to reduce the 30 parameters of the Larter-Breakspear model to a smaller subset of parameters that is more tractable for parameter inference (for a list of chosen parameters and reasons for inclusion/exclusion see S1 Table). We assessed 19 parameters as unlikely to be sensitive to common biological variables of interest and fixed them to default values. All default values were taken from [32]. The remaining eleven input parameters of the model were drawn from biologically relevant ranges (Table 3) using a Sobol sequence. The model was simulated with input parameters still on their original scales. Later, all eleven input parameters were scaled to within a range of [0, 1] for training the neural network (S9 Fig). Simulations were performed using Neuroblox. jl, a Julia library optimized for high-performance computing of dynamical brain circuit models (http://www.neuroblox.org); while the Larter-Breakspear model is technically unitless, the parameters are scaled so that a single timestep is 1 ms. [26]. Simulated time series were converted to BOLD using the Balloon-Windkessel model [58] ($T_R$ = 0.8s) [60] and then bandpass filtered ($0.01 < f < 0.1$ $Hz$) [61] to quantify functional connectivity [33]. We computed functional connectivity among the 78 brain regions from the processed time series and retained values above the diagonal to discard duplicates. The resultant 3,003 Pearson correlation values per sample constituted the output space for the neural network (S10 Fig). Simulated data were subject to two exclusion criteria to ensure biologically realistic behavior: time series with non-oscillatory behavior or with a mean FC larger than 0.3 were discarded. As a result, there were 4,730 retained samples for training and 526 for validation.

**Table 3. Investigated parameter subset of the Larter-Breakspear multi-scale brain model.**

| Parameter | Description | Range |
| --- | --- | --- |
| c | Global region-to-region coupling constant | [0.2, 0.5] |
| $\delta$ | Variance of thresholds | [0.64, 0.7] |
| $g_{Ca}$ | $Ca^{2+}$ conductance | [0.95, 1.05] |
| $V_{Ca}$ | $Ca^{2+}$ reversal potential | [0.95, 1.01] |
| $g_K$ | $K^+$ conductance | [1.95, 2.05] |
| $V_K$ | $K^+$ reversal potential | [−0.75, −0.65] |
| $g_{Na}$ | $Na^+$ conductance | [6.6, 6.8] |
| $V_{Na}$ | $Na^+$ reversal potential | [0.48, 0.58] |
| $a_{ee}$ | Excitatory-to-excitatory synaptic strength | [0.33, 0.39] |
| $a_{ei}$ | Excitatory-to-inhibitory synaptic strength | [1.95, 2.05] |
| $r_{NMDA}$ | NMDA/AMPA receptor ratio | [0.20, 0.30] |

### 5.3 Neural network

**5.3.1 Kepler orbit model.** We utilized a network with a fully connected architecture and a bottleneck layer in the middle to enforce compression. The network structure was determined empirically based on a trade-off between model accuracy and training speed at $k$ = 4 nodes in the bottleneck, equal to the number of original parameters. This approach was consistently applied in all our examples in this work. The selected architecture comprised two hidden layers before and two after the bottleneck layer, all of which had a *tanh* activation function. The final output layer, by contrast, was given a linear activation function ($f(x) = x$) to map to the output space. By the universal approximation theory of neural networks, hidden layers before and after the bottleneck layer had 14 and 110 nodes [62, 63]. The neural network was implemented with TensorFlow (version 2.6.0 [64]). For exact details of the applied architecture, see S11 Fig.

**5.3.2 $\beta IG$ model of glucose-insulin regulation.** Analogously, we applied a fully connected neural network to the glucose regulation model. The encoder comprised two hidden layers, each containing 50 nodes, while the decoder consisted of three hidden layers with 150, 300, and 300 nodes, respectively. All hidden layers were given rectified linear unit (*ReLU*) activation ($f(x) = \max(0, x)$), whereas the bottleneck and output layers were both implemented with a linear activation function. For further insight into the network architecture, please refer to S12 Fig.

**5.3.3 Multi-scale brain model.** A similar architecture was used for the brain model with minor adaptations to a significantly wider output space (3,003 values per sample). The encoder included two fully connected hidden layers with 21 nodes in each, whereas the decoder had a single hidden layer with 3,013 nodes. All hidden layers were given *ReLU* activation, whereas the bottleneck and output layers were both implemented with a linear activation function. Details of the network are described in S13 Fig.

### 5.4 Bottleneck analysis

To determine the number of uniquely resolvable latent parameters for a given model, we trained our neural network architecture on the previously described training data at varying bottleneck layer dimensions ($k \in \{1, 2, 3, \ldots, N\}$ where $N$ equals the number of original parameters) while keeping the rest of the network structure intact. At each $k$ increment, ten replicate training runs were performed independently. We employed 5,000 epochs during training to ensure model accuracy was not limited by training length. Both analyses involved batches of

256 samples. Model weights were updated by Adam optimizer [65] using mean squared error as the metric to optimize. We employed early stopping during optimization; we stopped training if validation accuracy had not improved for 200 subsequent epochs. From each replicate, we extracted the minimal validation error and the corresponding model weights. These were the outputs considered during subsequent analyses. For $k$ dimensions that were equal to or greater than the underlying complexity, validation error was expected to not decrease further with $k$. To distinguish the ideal $k$ from overparameterized solutions, we chose the smallest $k$ for which the error was not statistically significant from the minimum error across all $k$ values.

## 5.5 Global sensitivity analysis

Following the acquisition of a latent representation, we determined the influence of the original parameters on the latent parameters using global sensitivity analysis. We used the encoder (all layers before the bottleneck of the optimized neural network) to compute data pairs of input parameters and corresponding latent parameters. Since these pairs were unevenly sampled due to our filtering steps, we employed structural and correlative sensitivity analysis (SCSA), a method that handles non-uniform sampling and accounts for correlations among inputs, to compute global sensitivities [22]. SCSA partitions sensitivity into uncorrelated and correlated contributions. Of these, we used the uncorrelated component $S_{ij}^{unc}$, reflecting the exclusive contribution of each input parameter $x_i$, to more sparsely identify drivers of each latent variable $y_j$:

$$S_{i,j}^{unc} = \sum_{s=1}^{N} \left( f_{p_{i,j}}(x_i^{(s)}) \right)^2 / \sum_{s=1}^{N} \left( y_j^{(s)} - \bar{y}_j \right)^2 \tag{19}$$

Where $s$ is the sample index, $N$ is the total number of samples, and $f_{p_{i,j}}$ is a data-driven sensitivity function for $x_i$ with respect to $y_j$ [22]. We used SALib's (v1.4.5) implementation of SCSA [66, 67] to determine sensitivities. The above procedure was applied to every $j$-th latent parameter separately to derive each row of the $I \times J$ sensitivity matrix for $I$ input and $J$ latent parameters.

## 5.6 Global fitting

We employed global optimization to infer native and latent parameters from output data across all three model examples. In every instance, parameters were first normalized to a hypercube ($p_i \in [0, 1]$). For latent parameters, we determined bounds for the hypercube based on ranges that we observed for the latent parameters (S14, S15 and S16 Figs). Next, we used a basin-hopping algorithm (SciPy v1.7.1 [23]) to find the best-fitting parameters. Basin-hopping combines a global step-taking routine and a local optimizer to find a global minimum across the parameter space. The global step-taking routine was initialized at 0.5 within the hypercube, and each step involved a random displacement of coordinates with a step size of 0.2. Local minima were found at each step, including the initial point, using the Broyden–Fletcher–Goldfarb–Shanno (BFGS) algorithm. The goodness of fit was evaluated based on the residual sum of squares. Steps were accepted with a probability $P$ determined by the change in the value of the cost function $f(p)$.

$$P = exp\left(-\frac{f(x_{new}) - f(x_{old})}{T}\right) \tag{20}$$

Notice that the acceptance rate is 100% for steps that improve the objective but still nonzero for steps that yield worse objectives. This acceptance rate is tuned through a temperature

parameter $T$ individually adjusted for each model. This allows the algorithm to explore the landscape within the hypercube and thus greatly increases the likelihood of finding the global optimum.

## Supporting information

**S1 Fig. Summary of workflow with FixFit.**
(TIFF)

**S2 Fig. Kepler orbit model: Evolution of error during neural network training.**
(TIFF)

**S3 Fig. *βIG* model of glucose-insulin regulation: Evolution of error during neural network training.**
(TIFF)

**S4 Fig. Brain network model: Evolution of error during neural network training.**
(TIFF)

**S5 Fig. Kepler orbit model: Distributions of input parameters in the training dataset.**
(TIFF)

**S6 Fig. Kepler orbit model: Examples of model outputs from the training dataset.**
(TIFF)

**S7 Fig. *βIG* model of glucose-insulin regulation: Distributions of input parameters in the training dataset.**
(TIFF)

**S8 Fig. *βIG* model of glucose-insulin regulation: Examples of model outputs from the training dataset.**
(TIFF)

**S9 Fig. Brain network model: Distributions of input parameters in the training dataset.**
(TIFF)

**S10 Fig. Brain network model: Examples of model outputs from the training dataset.**
(TIFF)

**S11 Fig. Kepler orbit model: Neural network architecture.**
(TIFF)

**S12 Fig. *βIG* model of glucose-insulin regulation: Neural network architecture.**
(TIFF)

**S13 Fig. Brain network model: Neural network architecture.**
(TIFF)

**S14 Fig. Kepler orbit model: Distribution of latent parameters following training.**
(TIFF)

**S15 Fig. *βIG* model of glucose-insulin regulation: Distribution of latent parameters following training.**
(TIFF)

**S16 Fig. Brain network model: Distribution of latent parameters following training.**
(TIFF)

**S1 Table. Parameters of the Larter-Breakspear model with inclusion/exclusion criteria for this study.**
(PDF)

## Acknowledgments

We would like to thank David Hofmann for valuable discussions.

## Author Contributions

**Conceptualization:** Botond B. Antal, Helmut H. Strey, Lilianne R. Mujica-Parodi, Corey Weistuch.

**Data curation:** Botond B. Antal.

**Formal analysis:** Botond B. Antal.

**Funding acquisition:** Lilianne R. Mujica-Parodi, Corey Weistuch.

**Investigation:** Botond B. Antal, Anthony G. Chesebro, Corey Weistuch.

**Methodology:** Botond B. Antal, Helmut H. Strey, Corey Weistuch.

**Project administration:** Lilianne R. Mujica-Parodi, Corey Weistuch.

**Resources:** Lilianne R. Mujica-Parodi, Corey Weistuch.

**Software:** Botond B. Antal, Lilianne R. Mujica-Parodi, Corey Weistuch.

**Supervision:** Helmut H. Strey, Lilianne R. Mujica-Parodi, Corey Weistuch.

**Validation:** Botond B. Antal, Corey Weistuch.

**Visualization:** Botond B. Antal.

**Writing – original draft:** Botond B. Antal, Anthony G. Chesebro, Helmut H. Strey, Lilianne R. Mujica-Parodi, Corey Weistuch.

**Writing – review & editing:** Botond B. Antal, Anthony G. Chesebro, Helmut H. Strey, Lilianne R. Mujica-Parodi, Corey Weistuch.

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
