## [Decision Letter · Decision Letter 0]

17 Jan 2024

Dear Weistuch,

Thank you very much for submitting your manuscript "Achieving Occam’s razor: Deep learning for optimal model reduction" for consideration at PLOS Computational Biology.

As with all papers reviewed by the journal, your manuscript was reviewed by members of the editorial board and by several independent reviewers. In light of the reviews (below this email), we would like to invite the resubmission of a significantly-revised version that takes into account the reviewers' comments.

***

As you will see, both reviewers agree on the novelty and potential interest of your approach. However, they both raise similar concerns regarding the evidence you afford in favor of the breadth, efficiency and robustness of the method. They also demand that expose FixFit with much more mathematical details, and showcase it on real data. I believe these constructive criticisms highlight a clear opportunity for strengthening your work. Please add in your revised manuscript all the related analyses!

***

We cannot make any decision about publication until we have seen the revised manuscript and your response to the reviewers' comments. Your revised manuscript is also likely to be sent to reviewers for further evaluation.

Sincerely,

Jean Daunizeau

Academic Editor

PLOS Computational Biology

Marieke van Vugt

Section Editor

PLOS Computational Biology

Reviewer's Responses to Questions

**Comments to the Authors:**

Reviewer #1: Antal et al propose a novel use of deep learning for parameter identification in nonlinear dynamical systems. This is certainly an intriguing proposal and the paper plus proof-of-principle illustrations are well presented. However I feel the core algorithm needs to be more deeply exposed, broader in silico validation should be presented; and comparison with existing techniques should be performed and not just discussed. These issues are outlined in further detail below. If addressed, I think the paper could make an interesting and useful contribution to PLoS CB.

1. The overall architecture of the inference method is well presented in Figure but not the details evaded me. I assume that forward in silico simulations of the generative model over a range of parameter values are performed, and the timeseries from each run are fed through the neural network. I assume the neural network knows nothing about the generative model but rather (using the black box mechanism of multilayer NN’s) just learns the dimension == bottleneck dimension across the “landscape” of simulations. Some actual realizations of the technique in supplementary figures would be helpful. What about the observation function (e.g. the BOLD convolution in the neuronal/functional connectivity example)?

2. I’m not sure that 2 noise free examples are sufficient; a jump from a simple low dimension mechanical system to a complex high dimensional system. What about forward modelling of a bilinear or even first order nonlinear model (such as a 2 node nonlinear DCM [1])? And what is the role of observation noise, especially if auto-correlated – these are the sorts of challenging counterexamples of classic nonlinear time series methods stumbled through very poorly [2].

3. The conceptual comparison with Bayesian methods on p11 was very interesting but I think a 1 to 1 actual comparison is required. For example, using variational Bayes with Free Energy for model comparison can deal with some of the problems identified here, including conditional dependences between system parameters. By including an observation model with measurement noise priors, these approaches have been shown to work in the sort of deeper in silico validations I think are required here (see [1,3-5] for example).

References:

1. Stephan, K. E., Kasper, L., Harrison, L. M., Daunizeau, J., den Ouden, H. E., Breakspear, M., & Friston, K. J. (2008). Nonlinear dynamic causal models for fMRI. Neuroimage, 42(2), 649-662.

2. Rosso, O. A., Larrondo, H. A., Martin, M. T., Plastino, A., & Fuentes, M. A. (2007). Distinguishing noise from chaos. Physical review letters, 99(15), 154102.

3. Razi, A., Kahan, J., Rees, G., & Friston, K. J. (2015). Construct validation of a DCM for resting state fMRI. Neuroimage, 106, 1-14.

4. Daunizeau, J., Lemieux, L., Vaudano, A. E., Friston, K. J., & Stephan, K. E. (2013). An electrophysiological validation of stochastic DCM for fMRI. Frontiers in computational neuroscience, 6, 103.

5. Friston, K. J., Bastos, A., Litvak, V., Stephan, K. E., Fries, P., & Moran, R. J. (2012). DCM for complex-valued data: cross-spectra, coherence and phase-delays. Neuroimage, 59(1), 439-455.

Reviewer #2: In their paper “Achieving Occam’s razor: Deep learning for optimal model reduction” , Antal and colleagues present “FixFit” a method that uses simulation of dynamical systems together with bottleneck DNN (similar to autoencoders but mapping from parameters to data/timecourses instead of data to data). By finding the smallest bottleneck dimension that gives a validation accuracy close to the best, they can determine the dimensionality of the system simulations. The bottleneck layer of the DNN automatically defines a set of latent parameters. Once the network is learned, it can be used to fit new data and the authors can inspect the sensitive of latent parameters with respect to the original parameters of the system. The authors demonstrate the dimensionality reduction on to systems, a “simple” Kepler orbit equation and a higher dimensional dynamical system of neural activity.

This paper introduces an interesting method that could be very useful for many subfields of neuroscience, where models are often complicated, and it is not trivial to understand dependencies among parameters. Here, FixFit provides a method to analyse the model and its output and find the "minimal" amount of parameters needed to span the model output. However, the dimensionality reduction is based on simulated data and thus does not necessarily reflect the dimensionality of acquired data. In other words, Occam's razor, does not depend on the data. In that sense, an application to real data is missing. The paper is well written, although I think that the methods, in particular, the description of global fitting, would profit from more detail.

Overall, my assessment is very positive, but I think that the paper would profit from an illustration on real data and from some simulation that underlines the claims about noise. See my comments below.

Jakob Heinzle

Major:

Occam’s razor for real data?: While you illustrate how you can use FixFit to recover the minimal dimensionality needed to (re)cover simulations drawn from the entire parameter space (with some restrictions, see below), it is not clear to me how you would apply Occam’s razor on real data. You could create a Bottleneck network and then fit to real data. It would be really good, if you could show this. It should not be difficult to get these data, e.g. for the resting state FC. However, how would you determine then, whether the model has the right dimensionality for the real data? Would you need to retrain your bottleneck network on real dat? One could also think of this as the following question: Is there a way of assuring that your latent space dimensionality is the dimensionality of what your model can explain in the real data? Could that dimensionality be even lower? Even if you are not able to solve this, it would be good to discuss it and to show an application to real data.

Data fitting: It was not clear to me how the “data” fitting procedure was performed. I understand that this is done at then end, once a bottleneck network is trained and validated. From this network you then try to infer the latent states and parameters. Which data were used for the global fitting analysis? New simulations, or one of the data points used for training or validation of the DNN. Are the points displayed in figures 2C and 2D different data points or different solutions when fitting one single simulated orbit? I assume it is one orbit. Why did the optimizer evaluation stop at solutions that are not fitting well? Or what do you mean by optimizer evaluation?

As said, I would assume that global fitting happens on a learned network. You say, that the algorithm you apply does not guarantee to find the global optimum. Could you elaborate on this? Does this imply that it is still very difficult to find the best and, therefore, the unique solution, which you want to find (according to the abstract). One of the reasons for reducing the parameter dimensionality is to avoid problems for fitting when the landscape can show flat ridges etc due to correlated parameters. Please comment on this.

Noise: In the introduction you write about how FixFit could be used “to gauge how resolvable complexity depends on the amount of noise.” I think this is very interesting, but I was missing any consideration of noise in your results/simulations. It was not clear to me, how you would actually do this. I think it would be good to illustrate this gauge effect in a simulation/application. Is the bottleneck embedding sensitive to noise, or not? How can applying FixFit only on simulated data (without noise) help with this gauge?

Minor:

You present two very specific bottleneck-network architectures, presumably optimized for the two example cases (dynamical systems). How to choose the correct/best network with bottleneck? Does it matter which bottleneck network one choses? I think it would be good to at least briefly discuss this.

Heuristic to determine bottleneck size: You define the bottleneck dimension as the lowest dimension with a fit not significantly worse than the best. In the two cases you present, this dimension is very clear and also observable as a knee. Would you expect this always to be the case (also for other models) and to what degree could noise flatten the curve and thus make it more difficult to define the bottleneck dimension. Could you give the reader some hint/indication towards this problem?

Fixing parameters in the multi-scale model (line 272ff): You fix 19 parameters that you ”determine to be unlikely to be sensitive”. Who did you determine those? Wouldn’t it be exactly the purpose of FixFit to actually determine these parameters?

Line 66/67: How is Figure S2 related to global optimization and unique solutions? It shows the distribution of latent parameters. Please clarify.

Supplementary Figures S10 and S11: Please provide more information in the caption for readers who are not familiar with the notation you have chosen. I understand that the two lowest lines are input and output dimensions. Is this correct. It would also be nice, if you could highlight the bottleneck layer.

Please specify what a ReLu activation is.

**Have the authors made all data and (if applicable) computational code underlying the findings in their manuscript fully available?**

Reviewer #1: Yes

Reviewer #2: Yes

PLOS authors have the option to publish the peer review history of their article (what does this mean?). If published, this will include your full peer review and any attached files.

Reviewer #1: **Yes: **Michael Breakspear

Reviewer #2: **Yes: **Jakob Heinzle

Figure Files:

Data Requirements:

Reproducibility:

To enhance the reproducibility of your results, we recommend that you deposit your laboratory protocols in protocols.io, where a protocol can be assigned its own identifier (DOI) such that it can be cited independently in the future. Additionally, PLOS ONE offers an option to publish peer-reviewed clinical study protocols. Read more information on sharing protocols at <a href="https://plos.org/protocols?utm_medium=editorial-email&utm_source=authorletters&utm_campaign=protocols">https://plos.org/protocols?utm"

---

## [Decision Letter · Decision Letter 1]

27 Jun 2024

Dear Weistuch,

We are pleased to inform you that your manuscript 'Achieving Occam’s razor: Deep learning for optimal model reduction' has been provisionally accepted for publication in PLOS Computational Biology.

Best regards,

Marieke Karlijn van Vugt, PhD

Section Editor

PLOS Computational Biology

Marieke van Vugt

Section Editor

PLOS Computational Biology

Dear authors,

Congratulations! Both reviewers have signed off on your manuscript. Reviewer 2 makes a minor suggestion, which I think would be good to incorporate in the final version of your manuscript.

Sincerely,

Marieke

Reviewer's Responses to Questions

**Comments to the Authors:**

Reviewer #1: I appreciate the detailed response to the prior review and the additional analyses that addressed the outstanding issues

Reviewer #2: In this revised submission, Antal and colleagues have incorporated several changes that clearly improved the paper. I have one final point regarding the application to real data (blood glucose-insulin regulation). Other than that, all my concerns are addressed.

Jakob Heinzle

Application to real data: There are two things I was missing when reading your newly included analysis. First, it was not clear how the fitting of the latent representation to experimental data was performed. This is mentioned only very briefly and I could not follow how you did this. Second and more importantly, I think you should discuss very clearly that your latent representation is based on the model, not on the data. This is crucial: In the title you mention Occam’s razor. Occam’s razor roughly states that the explanation (model) should be as simple as possible given the thing that needs to be explained (data). In your case, the data has no impact at all on the complexity of the model. FixFit is a tool to analyse/optimize models (through parameter-compression). This is a limitation of FixFit compared to what one would expect from Occam’s razor and should be openly discussed in the paper.

As a side note: The title on the FixFit homepage on lcneuro.org describes the method perfectly.

**Have the authors made all data and (if applicable) computational code underlying the findings in their manuscript fully available?**

Reviewer #1: Yes

Reviewer #2: Yes

PLOS authors have the option to publish the peer review history of their article (what does this mean?). If published, this will include your full peer review and any attached files.

Reviewer #1: **Yes: **Michael Breakspear

Reviewer #2: **Yes: **Jakob Heinzle

---

## [Editor Report · Acceptance letter]

11 Jul 2024

PCOMPBIOL-D-23-01512R1 

Achieving Occam’s razor: Deep learning for optimal model reduction

Dear Dr Weistuch,

I am pleased to inform you that your manuscript has been formally accepted for publication in PLOS Computational Biology. Your manuscript is now with our production department and you will be notified of the publication date in due course.

With kind regards,

Zsofia Freund
